# OpenReview forum: "FEWL: Measuring and Mitigating LLM Hallucination Without Gold-Standard Answers"
_ICLR.cc/2026/Conference — ICLR 2026 Conference Withdrawn Submission_

### Official Review · Reviewer_vdaX · 2025-10-16

**Soundness:** 3
**Presentation:** 3
**Contribution:** 3
**Rating:** 6
**Confidence:** 5

**Summary:**

The paper introduces FEWL (Factualness Evaluations via Weighting LLMs), a metric to measure and mitigate LLM hallucinations without gold‑standard answers. FEWL uses a panel of reference LLMs and assigns question‑dependent expertise weights by contrasting each reference model’s agreement with Intentionally Wrong (IW) answers versus their Corrected (CO) negations. It then combines (i) an expertise‑weighted truthfulness term—semantic similarity between the evaluated answer and the reference answers, reweighted by learned expertise—with (ii) a laziness penalty that down‑weights answers that also look similar to reference answers on nearest‑neighbor questions (to discourage superficial, topic‑level responses). The two parts are combined via a variational f‑divergence formulation, with a concrete implementation using total‑variation; the paper also explores JS and KL variants in the appendix. Theoretical analysis (Theorem 3.4) claims FEWL will rank the least hallucinating model highest in expectation, under assumptions including constant expertise and a conditional independence structure. Empirically, FEWL is evaluated on CHALE (940 Qs), TruthfulQA (737 Qs), and HaluEval (10k Qs) for both measurement (distinguish non‑hallucinated vs hallucinated answers) and ranking (model‑ and sample‑level). The approach outperforms baselines such as single reference with/without laziness, and uniform multi‑reference without expertise weighting. The authors further show mitigation use cases: FEWL‑guided in‑context learning (ICL) selection and label‑free SFT, both improving judged quality vs vanilla baselines. A notable practical claim is cost: $<0.3 per 1k evaluations (vs human annotation $16/h) and about $0.2 to evaluate the full TruthfulQA with GPT‑3.5 references, partially attributed to batched generation of IW/CO and efficient similarity (Vectara HHEM). [13019_FEWL..._and_Mitig]

**Strengths:**

Clear, practical problem focus (no gold answers)
Many labs lack high‑quality gold answers; FEWL’s design explicitly targets this gap and gives a continuous hallucination score. The practicality is underlined by cost estimates and parallelizable steps.

Expertise estimation without ground truth
Using disagreement with wrong answers (IW vs CO) to infer expertise is intuitive and empirically effective. The normalization via softmax (λᵢ) is simple and principled, and ablations (uniform vs FEWL vs “ideal” λ) support the importance of the weighting.

Laziness penalty idea and justification
Penalizing answers that resemble those given to similar but distinct questions is a thoughtful way to discourage superficial pattern‑matching; Figure 2 and Appendix experiments provide statistical justification that similar questions often need different correct answers.

Theoretical framing and guarantee
The variational f‑divergence view is elegant; Theorem 3.4 gives an expectation‑level guarantee that the best model receives the highest FEWL score under stated assumptions. This is a welcome attempt at providing measurable conditions for a no‑gold setting.

Broad and multi‑granularity evaluation
Results span three datasets (CHALE, TruthfulQA, HaluEval), include pairwise ranking, model‑level ranking, and show mitigation via ICL and label‑free SFT with concrete win‑rate results (GPT‑4 and human judges where available). The appendices include ablation on λ, similarity variants (HHEM‑2.1 vs 1.0), different f‑divergences, KNN choices, and timing.

Reproducibility‑helpful details
The paper provides prompts for IW/CO generation, default R=25 IW/CO pairs, K=25 neighbors, similarity model details (Vectara HHEM), and some thresholds for neighbor selection, which will help practitioners replicate and adapt.

**Weaknesses:**

Reliance on reference LLM quality & assumptions
The approach presumes that at least one reference model has sufficient expertise on a question. The authors acknowledge feasibility issues if none does. Moreover, the constant expertise (Assumption 3.1) and conditional independence (Assumption 3.3) are strong and may be violated in realistic multi‑domain settings (expertise can vary sharply by topic and references often share correlated failure modes). The empirical support in D.5 is helpful but still limited.

Similarity‑model dependency and bias
FEWL’s core comparisons rely on a specific semantic similarity model (Vectara HHEM). The risk is that any idiosyncrasy or bias of this model propagates into FEWL scores (e.g., stylistic similarity rewarded over factual grounding). Although Appendix D.8 explores versions 1.0 vs 2.1 and trends hold, broader cross‑metric corroboration (e.g., different embedding/STS or entailment models) would improve confidence.

Laziness penalty could over‑penalize legitimate overlap
The penalty is triggered when an answer resembles reference answers to neighbor questions. Some domains legitimately share canonical statements (e.g., definitions, constants, safety disclaimers). Despite the paper’s thresholding to avoid overly similar neighbors, the method may still down‑weight correct but general facts, especially in knowledge areas with stock formulations. A more semantic notion of task‑relevance vs topic overlap might be needed.

Use of closed models and judge‑LMs raises reproducibility and circularity concerns
Several evaluations use GPT‑4 as a judge and reference; the text also mentions GPT‑5 as a baseline in tables and in the “Use of LLMs” section (for readability), which may be problematic for strict reproducibility (availability, version drift) and double‑blind considerations. It would help to include strong open‑model replications and human‑only judgments for key claims.

Scope of evaluation tasks
The benchmarks are QA‑centric and primarily short‑form. Hallucination manifests differently in long‑form, tool‑augmented, or domain‑specific reasoning tasks (math proofs, finance, medical). It’s unclear how FEWL behaves when reference models are brittle on symbolic or retrieval‑heavy problems.

Potential gaming & stability
Because FEWL relies on (i) a particular IW/CO generation prompt and (ii) a specific similarity model, one could optimize model outputs to align with that similarity space rather than factuality per se. The paper does not analyze adversarial references (e.g., deliberately low‑quality panels) or prompt sensitivity/variance (temperature, seeds) in depth.

Statistical reporting
Many tables show percentage/counts improvements, but formal significance testing and confidence intervals are not always present for main‑text results (some ± shown in later ablations). Stronger statistical treatment would bolster claims of consistency.

Complexity vs baseline simplicity
While the per‑sample cost is modest, the full pipeline has multiple moving parts (IW/CO generation, expertise estimation, KNN search, divergence aggregation). A head‑to‑head with self‑consistency, multi‑judge voting, or retrieval‑augmented verifiers—all without gold—would help position FEWL among simpler alternatives.

**Questions:**

Reference panel composition & robustness
How sensitive is FEWL to the mix and number of reference LLMs? Can you report performance as a function of N (e.g., 1, 3, 5, 8 references) and under adversarial or weak references (e.g., all small/open models)? What’s the minimal competent panel for stable estimates?

Assumption stress tests
Can you empirically test violations of Assumption 3.1 (expertise varying per topic) and 3.3 (dependence between references and candidate model due to training coincidences), and quantify how FEWL’s ranking guarantee degrades?

Laziness penalty calibration
How do you set the neighbor similarity thresholds and K in practice to avoid penalizing legitimate general statements? Could you replace raw similarity with entailment‑conditioned overlap or topic disentanglement to better capture “superficiality” vs “shared truths”? An ablation with purely definitional questions would be helpful.

Similarity model dependence
Beyond HHEM 1.0/2.1, can you replicate with multiple independent similarity/entailment models (e.g., SBERT, DeBERTa‑MNLI, E5, GritLM) and report agreement/variance? A model‑agnostic ensemble might reduce metric bias.

Prompt and seed sensitivity
The IW/CO generation quality matters. How stable are results w.r.t. different IW prompts, temperatures, and seeds? Could you add a diversity‑control mechanism (e.g., nucleus sampling constraints) and quantify trade‑offs?

Mitigation pipeline choices
For FEWL‑guided ICL and label‑free SFT, did you study how data selection varies the downstream gains (e.g., top‑k by FEWL vs random vs uncertainty sampling), and how sensitive SFT is to noise in selected answers? Also, can you report human‑only evaluation for ICL/SFT in addition to GPT‑4 judges?

Generalization beyond short‑form QA
Have you tried FEWL on long‑form factual generation, citation‑grounded tasks, or reasoning with tools? Does the laziness penalty still help when neighbor questions are multi‑step or when answers must include sources?

Reproducibility artifacts
Will you release code, prompts, IW/CO pools, and reference‑answer caches (where licensing allows)? Given mentions of GPT‑5 in tables and “Use of LLMs,” how will you guarantee reproducible runs for reviewers using only publicly accessible models?

Metric interpretability
Can you expose per‑component attributions (e.g., which reference models and which neighbors drove a low/high score) to help practitioners diagnose failures and improve their systems?

Potential for gaming
What safeguards or adversarial tests can detect models trained to optimize FEWL’s similarity space rather than factuality (e.g., templated phrasing that maximizes HHEM similarity while being false)? Could cross‑metric ensembles mitigate this?

---

### Official Review · Reviewer_ncPT · 2025-10-30

**Soundness:** 2
**Presentation:** 3
**Contribution:** 3
**Rating:** 4
**Confidence:** 3

**Summary:**

This paper studies the evaluation of LLM factuality without gold-standard answers. The authors propose a hallucination metric named Factualness Evaluations via Weighting LLMs (FEWL), which weights the expertise from multiple reference LLMs to form a proxy ground truth. Theoretical analysis shows that FEWL will in expectation assign the highest score to the best-performing model regardless of whether the gold answer is provided. Experiments show that FEWL can be used to reduce hallucination via in-context learning or supervised fine-tuning.

**Strengths:**

- The paper considered a novel setting of hallucination evaluation without gold answers.
- Provided theoretical analysis on the validity of the proposed FEWL metric.
- Conducted evaluation across three standard benchmark datasets.

**Weaknesses:**

- The core mechanism of the proposed method seems to be a sophisticated form of consensus-seeking among reference LLMs. Equating consensus to factuality can create a risk, e.g., if all reference models share a common well-established misconception, FEWL may mis-identify this as "expertise" and assign a high score to the popular but false answer.

- The theoretical guarantee is based on the simplifying assumption that an LLM's expertise is constant across all questions. This directly contradicts Algorithm 1, where the expertise score is estimated on a per-question basis, and thus does not fully support the method as implemented.

- Another concern is that the expertise-weighting component might be entirely dependent on the quality of the generated "Intentionally Wrong" and "Corrected" answers. If this generation step produces answers that are trivially wrong, or conversely, too subtly incorrect, the resulting expertise scores may be arbitrary and unreliable.

- The laziness penalty operates on the assumption that correct answers to textually similar questions should be different. This heuristic is fragile and can fail in real-world scenarios where different (but similar) questions share the same answer.

- While the method is cheaper than human annotation, it may introduce non-trivial computational overheads.

**Questions:**

- We observe from Table 1 that the "single + penalty" baseline often performs nearly as well as the full FEWL method, particularly when using GPT-5 as a reference. Furthermore, the performance difference between using nearest-neighbor questions and random questions for the laziness penalty appears marginal (as seen in Table 11). Could the authors discuss the implications of these findings?

- Could the authors clarify whether approximate KNN or exact KNN was used in the implementation?

---

### Official Review · Reviewer_jLLD · 2025-11-01

**Soundness:** 2
**Presentation:** 1
**Contribution:** 2
**Rating:** 4
**Confidence:** 5

**Summary:**

This paper introduces FEWL (Factualness Evaluations via Weighting LLMs), a framework for measuring and mitigating hallucination in large language models (LLMs). FEWL leverages off-the-shelf reference LLMs to generate proxy answers, quantifying each reference LLM's expertise through intentional wrong answers and corrected versions, combined with a laziness penalty to penalize superficial responses. Empirically, FEWL is evaluated on datasets like Truthful-QA, CHALE, and HaluEval, showing improved hallucination measurement accuracy compared to naive baselines.

**Strengths:**

1. The paper targets a critical and underexplored problem in LLM reliability. That is, reducing hallucinations of LLMs without relying on expensive human annotations.

2. The paper is easy to understand.

The paper is logically expressed, and the description of the method is complete and intuitive. The authors explain the principles and details of the FEWL with pictures and examples, which makes it easy to understand.

**Weaknesses:**

1. Lack of Evidence for Key Assumptions

The soundness of the proposed method is somewhat undermined by the absence of concrete evidence for several critical assumptions. The REFERENCE LLM EXPERTISE WEIGHTING module hinges on the assertion that "the ability of an LLM to discern an apparently wrong answer strongly correlates with the expertise." However, the authors do not provide the reference of this assumption or validate it. It appears to be based on the authors' observations, yet the details of these observations are not elaborated, casting doubt on the reliability of the conjecture.

Similarly, the LAZINESS PENALTY module is built on the premise that "expert LLMs are unlikely to give similar answers to different questions, even though questions are regarding the similar topic." While the authors attempt an Empirical Justification, the experimental description lacks clarity. For instance, the use of GPT-4 to judge the consistency between questions and answers might introduce GPT-4 hallucinations that affect the results. Moreover, the validation is only conducted on the Truthful-QA dataset, which has a limited sample size. Due to the artificial nature of the dataset construction, the same answers are often restricted in their frequency of appearance to avoid repetition. Without ruling out dataset design effects, the justification does not convincingly support the assumption.


2. Incomplete and Flawed Experiments

The experiments presented in the paper lack key details and suffer from several flaws. For example, the specific LLMs used as reference LLMs in the experiments are not clearly specified.
Additionally, the paper does not include essential ablation experiments. Given the importance of reference LLMs to the FEWL method, the authors fail to explore how to select reference LLMs and the optimal number of them.

Similarly, the computation of similarity scores is vital to the FEWL method, yet the authors simply use Vectara for this purpose without explaining the rationale behind this choice or comparing it with other traditional similarity score calculation methods. Instead, they only compare different versions of Vectara.

Furthermore, the paper lacks comparisons with baseline methods. All the experiments conducted are ablation studies and do not involve comparisons with existing methods. Despite the authors' claim that this work pioneers the study of "hallucination measurement without gold-standard answers," it is still necessary to compare the proposed method with existing hallucination detection methods, many of which can also detect hallucinations without gold-standard answers.

Finally, the experimental design statement "We use multiple answers given by a single reference LLM as the set of reference answers and instruct a single reference LLM to generate multiple diversified answers instead of leveraging each single answer from multiple reference LLMs, for time efficiency" is perplexing. This experimental setup seems to deviate from the method's design, making the experimental results seem unrepresentative of the FEWL method's effectiveness.

3. Presentation and Technical Errors

There are several presentation and technical issues.
At line 209, “answer x” should be “question x.”
Indexing/encoding problems appear at lines 862 and 928 (garbled text).
Most of the citations throughout the paper also seem to be problematic, with Author-Date references directly appearing in the text without being enclosed in parentheses.

**Questions:**

1. Could the authors provide more concrete evidence or validation for the key assumptions underpinning the REFERENCE LLM EXPERTISE WEIGHTING and LAZINESS PENALTY modules?

2. How were the reference LLMs selected for the experiments, and what criteria were used to determine their suitability?

3. Why was Vectara chosen for the computation of similarity scores, and how does it compare to other traditional similarity score calculation methods?

4. Could the authors also explain the reasoning behind using a single reference LLM to generate multiple diversified answers instead of leveraging multiple reference LLMs?

5. Why were no comparisons made with existing hallucination detection methods that also operate without gold-standard answers?

---

### Official Review · Reviewer_Qpmi · 2025-11-01

**Soundness:** 2
**Presentation:** 2
**Contribution:** 3
**Rating:** 4
**Confidence:** 3

**Summary:**

This paper proposes FEWL (Factualness Evaluations via Weighting LLMs), a novel hallucination metric designed to evaluate LLM outputs without requiring gold-standard answers. The key innovation is leveraging multiple reference LLMs as proxies, with two main components: (1) expertise weighting based on how well each reference LLM distinguishes intentionally wrong (IW) answers from corrected (CO) answers, and (2) a "laziness penalty" that penalizes LLMs giving similar answers to semantically similar questions. The authors provide theoretical guarantees and demonstrate FEWL's effectiveness on TruthfulQA, CHALE, and HaluEval datasets, showing applications in both in-context learning and supervised fine-tuning.

**Strengths:**

- Addresses the real cost and error-proneness of human annotation
- Three datasets (TruthfulQA, CHALE, HaluEval) with multiple metrics
- Shows both measurement and mitigation (ICL + SFT)
- Tables 12-13 provide good ablation of the λ_i component
- Effort to provide theoretical justification, even if imperfect

**Weaknesses:**

- Constant expertise (Assumption 3.1) contradicts the problem motivation; conditional independence (Assumption 3.3) is not empirically validated
- Using LLM-generated IW/CO answers to evaluate LLMs doesn't truly eliminate the need for ground truth—it just shifts the problem
Limited theoretical insight: The f-divergence connection (Proposition C.1) provides bounds but doesn't give clear intuition about when/why - Figure 2 shows similar questions have different answers, but this doesn't directly validate that answer similarity implies lower expertise. Need more direct validation.
- Missing comparisons:
1. No comparison with self-consistency methods (explicitly mentioned in related work)
2. No comparison with simple ensemble baselines (e.g., majority voting)
3. Limited analysis of failure cases


- Hyperparameter sensitivity:

1. Choice of K (neighbors), R (IW/CO pairs) seems arbitrary (fixed at 25)
2. Similarity threshold for KNN not well justified
3. Temperature in softmax for λ_i not discussed


- Reproducibility concerns:

1. Anonymous submission but mentions specific model versions
2. Prompts in Appendix but not clear if code will be released
3. Vectara model dependency

**Questions:**

- Can you provide empirical evidence that Assumption 3.1 holds in practice? The data in Figure 4 suggests expertise varies significantly.
- How does FEWL compare to SelfCheckGPT (Manakul et al., 2023) which also doesn't require gold labels?
- When does FEWL fail? Can you provide examples where all reference LLMs are wrong or where the laziness penalty hurts performance?
- How does FEWL perform on:

1. Specialized domains (medical, legal, scientific)?
2. Non-English languages?
3. Long-form generation (beyond QA)?


- How many reference LLMs are needed? What happens if you use 10 vs 3 vs 1?
- Can you provide detailed timing analysis comparing FEWL vs traditional evaluation with human labels (including data collection time)?
- In Theorem 3.4, how loose is the bound in practice? Can you quantify the gap between FEWL score and true hallucination rate?

---

### Note · Authors · 2026-01-05

I have read and agree with the venue's withdrawal policy on behalf of myself and my co-authors.